# Dynamics of Soil Organic Carbon and CO$_2$ Flux under Cover Crop and No-Till Management in Soybean Cropping Systems of the Mid-South (USA)

Alexandra G. Firth [1,*], John P. Brooks [2], Martin A. Locke [3], Dana J. Morin [1], Ashli Brown [4] and Beth H. Baker [1]

1 Department of Wildlife, Fisheries and Aquaculture, Mississippi State University, Mississippi State, MS 39762, USA
2 USDA-ARS, Genetics and Sustainable Agriculture Research Unit, Mississippi State, MS 39762, USA
3 USDA-ARS, National Sedimentation Laboratory, Oxford, MS 38655, USA
4 Department of Biochemistry, Molecular Biology Entomology and Plant Pathology, Mississippi State, MS 39762, USA
* Correspondence: agg224@msstate.edu

**Abstract:** The transition of natural landscapes to agricultural uses has resulted in severe loss of soil organic carbon, significantly contributing to CO$_2$ emissions and rising global temperatures. However, soil has the largest store of terrestrial carbon (C), a considerable sink and effective strategy for climate change mitigation if managed properly. Cover crops (CC) and no-till (NT) management are two management strategies that are known to increase percent organic carbon (%OC); however, adoption of these practices has been low in the mid-South due to lack of region-specific research and resistance to unproven practices. Therefore, the purpose of this study was to evaluate the impacts of CC-NT treatments in soybean cropping systems on soil percent organic carbon (%OC) and CO$_2$ flux following long-term implementation. Results showed significantly greater %OC in NT (1.27% ± 0.03) than reduced till (RT; 1.10% ± 0.03; $p < 0.001$) and greater in both CC (rye: 1.23% ± 0.03, rye + clover: 1.22% ± 0.03) than no cover (1.11% ± 0.03; $p < 0.001$). Bacterial abundance ($p = 0.005$) and pH ($p = 0.006$) were significant predictors of %OC. There was no overall significant difference in CO$_2$ flux between tillage or CC treatments; however, there were significant differences between NT and RT in July of 2020 when %RH increased ($p < 0.001$). Bacterial abundance negatively impacted CO$_2$ flux ($p < 0.05$), which contradicts most studies. The rate of proportional change and pattern of variability in C pools suggested loss of %OC in RT treatments that were not apparent when considering %OC alone. The results of this study provide valuable insight into C turnover and the effectiveness of CC use in the Mid-South to increase soil C stocks.

**Keywords:** carbon; no-till; cover crops; soil carbon sequestration; conservation agriculture; carbon measurement

## 1. Introduction

Land use changes from natural landscapes to agriculture and pasture lands account for 32% percent of carbon dioxide (CO$_2$) emissions globally [1]. Conventional agricultural practices can deplete historic soil carbon (C) stocks by as much as 60%, largely because of poor soil practices that reduce C without an equivalent C input, which ultimately lead to declines in soil health and productivity as well as increased C emissions into the atmosphere [2]. However, soil has the largest store of terrestrial C and the potential to store 0.4–5 Gt C/year, a considerable C sink and effective strategy for climate change mitigation if managed properly [3–5].

To that end, sequestering soil C has gained attention globally with the rise of C trading, which provides producers and landowners a market-based financial incentive to minimize

greenhouse gas emissions [6]. With an increased economic value placed on soil organic carbon, the implementation of agricultural practices that sequester C are critical for climate change mitigation and the financial well-being of the growers who invest in C sequestration strategies. Soil greenhouse gas (GHG) mitigation is dependent upon storing C in stable form, which is typically achieved by the adsorption of C compounds into mineral soil and low rates of microbial respiration [7]. However, building labile fractions of soil organic carbon are essential for effective nutrient cycling, soil biological activity and the creation of soil aggregates, which are all characteristics that contribute to soil health, but often result in increased soil respiration [8–10]. Thus, agriculture is dually challenged to promote C flow through the soil system for agronomic vitality and prioritize soil C persistence for global climate mitigation goals.

Cover crops (CC) and no-till (NT) are two conservation management practices that increase %OC and provide soil health benefits by increasing microbial diversity and activity [11], suppressing weeds, preventing erosion, and increasing moisture retention [12,13]. Under certain circumstances, and when CC are used in combination with no-till (NT) management, C in agriculture fields can increase by as much as 70% [14]. However, differences in CC species, management practice implementation, and regional climate create difficulties in predicting the effect of conservation practices on soil C stocks. For example, Zablotowicz et al. [15], in a study of no-till combined with CC, found this combination of treatments to have the greatest accumulation of total organic carbon but Chu et al. [16] found that soil organic carbon did not significantly differ across treatments and was comparable to baseline measurements. Additionally, Aldridge et al. [17] found that CC could only explain 5% of the variation in soil total carbon between treatments.

Most studies addressing changes in soil organic carbon following NT and CC management aim to detect differences between treatment groups, with the intent to determine the effectiveness of the management practice towards various agronomic goals and ecosystem services. While this framework is informative when addressing soil health, it is less helpful if the objective is to sequester C for use in a C trading system. For example, an increase in percent organic carbon (%OC) following CC application is usually accompanied by a simultaneous increase in $CO_2$ emissions [18–20]. Therefore, the observed increases in %OC do not directly translate into C stabilization, but are an indication increased soil health. Alternatively, processed-based models, like those used in C trading markets, use soil characteristics, climate, and management to predict C sequestration potential. This approach requires complex algorithms and many time measurements to place a value on tons of $CO_2$ mitigated from the atmosphere. These modeling approaches are the subject of much controversy, have considerable uncertainty, and do not illustrate incremental shifts in soil organic carbon over short time frames [21–23] and thus are impractical for producers designing an adaptive management plan which must be tailored and adjusted to site-nuisances.

These limitations become of particular concern in the mid-South, USA where a lack of research specific to regional nuisances has resulted low adoption rates of conservation management, despite National Resources Conservation Service (NRCS) financial assistance programs [24]. The Mississippi Alluvial Valley (MAV) region of the mid-South is characterized by intensive agriculture land use, historically alluvial soils, high rainfall (1375 mm/year) and above freezing winter temperatures (average high 16.1 °C, low 2.2 °C), all factors that promote rapid decomposition of plant material and $CO_2$ emissions via microbial respiration.

If CC and NT management are to be integrated into the mid-South, studies must directly address both differences in soil organic carbon between treatments and provide readily transferable information on changes to C pathways following management implementation. To address these limitations of site-specific C assessment, this study proposes a conceptual method whereby, measures of C pool turnover, via rate and proportional variability in C pools was assessed using multivariate measures [25], indicating whether shifts in C pool composition are directional (i.e., towards %OC accumulation or loss).

We describe site-specific changes in C pools relative to each other which can be rapidly interpreted and translated into action, either by management that continues to support C accumulation or by shifting strategies towards those that minimize C emissions.

Thus, this study had three purposes towards producer adoption of conservation management: (1) to assess the impacts of CC-NT treatments in soybean cropping systems on soil %OC and $CO_2$ Flux after several years of implementation, (2) assess the relationship of bacterial abundance, fungal abundance, soil moisture, soil temperature, cation exchange capacity (CEC), percent relative humidity (%RH) and pH on %OC and $CO_2$ Flux, respectively, and (3) model the directional change of site-specific C pools under different management. The results of this research provide valuable insight into soil C cycling and the effectiveness of CC use in the Mid-South to increase soil C stocks.

## 2. Materials and Methods

### 2.1. The Study Site and Design

The study was conducted in 2019 and 2020 at the USDA-ARS Crop Productions Systems Research Unit farm near Stoneville, Mississippi (MS), USA. In fall 2000, experimental plots were established under cotton (*Gossypium hirsutum* L.) production in a split-block design as described in Locke et al. [26]. Within each of four blocks, six treatment plots (n = 24) were constructed as 8 m wide and 32 m long (0.026 ha) with 8 1 m wide raised-bed rows in each plot. No-tillage or reduced tillage (NT or RT) was the main effect, with CC (abruzzi rye (*Secale cereale*), balansa clover (*Trifolium michelianum*) or none) as the split effect. Cotton was planted as the cash crop until 2007. Since spring of 2008 to present, Asgrow AG4632 variety soybean (*Glycine max*) was planted as the cash crop. The experimental area with continuous soybeans did not include a CC from 2008 to 2014, but tillage treatments from the previous experiment (RT or NT) were maintained. Cover crop treatments were added again in 2014 with elbon rye (*Secale cereale*), crimson clover (*Trifolium incarnatum*), or no cover [27,28]. In 2017 until present, the following cover crop treatments were established: elbon rye (RY), elbon rye+crimson clover mix (RC) and no cover (NC). Tillage treatments have remained the same since plot establishment in 2000. The combinations of treatments in the present study were as follows: NTxRY, NTxRC, NTxNC, RTxRY, RTxRC, and RTxNC. In late April of 2019 and 2020 when rye CC reached peak growth, the CC was terminated with two applications of Gramoxone SL2.0 (Syngenta, Greensboro, NC, USA) 2 weeks apart at a rate 0.77 kg ha$^{-1}$ and rolled for cash crop planting. Potash was applied to all treatment plots in January 2020.

### 2.2. Cover Crop Biomass and Decomposition Rate

Biomass was sampled by selecting 2 random locations within each plot and cutting all plant residue all vegetation within a 1 m$^2$ area. Volunteer vegetation was used for NC plots. The most common volunteer vegetation in NC plots was crimson clover (*Trifolium incarnatum*) and common purslane (*Portulaca oleracea*). Plant biomass from each location was dried at 105 °C and the dry mass weight was measured. Twenty-four hours after CC termination, 150 g of CC biomass was cut and placed in litterbags measuring 20.23 cm × 20.23 cm made from fiberglass mesh screen (0.508 mm mesh size) [29]. Two litter bags containing CC biomass (or volunteer vegetation) from their prospective plots were stapled closed and then secured on the soil surface (approximately mid-May). There was a total of 48 litter bags, two litter bags per plot. Litter bags were collected in September, dried and weighed. The weight of the litterbags from each plot was averaged ($W_e$) and then divided by the starting weight ($W_i$; 150 g) to obtain a decomposition rate. The rate was multiplied by the initial CC biomass ($CC_i$) to estimate remaining plant residue at the end of the season ($CC_e$) within each plot (Equation 1). The midpoint between the starting and final biomass was used as a proxy for mid-season CC biomass in the directional change model.

$$\frac{W_i}{W_e} \times CC_i = CC_e \tag{1}$$

## 2.3. Soil Sampling

Two soil samples per plot were taken, one from beneath each duplicate litter bag location with soil cores (3.8–4 cm diameter × 10 cm depth) collected at the date of litterbag implementation (approximately mid-May 2019 and 2020) and once a month through September of each year. Cores from the same plot were consolidated into one composite sample per plot. There were 24 samples per sampling event (5 sampling events, May–September), 120 samples per year, for a total of 240 samples over two years (2019 and 2020). The initial sampling event (in May after CC termination) occurred before litter bags were secured to the soil surface; all subsequently collected samples were taken from the soil directly under the litterbag by lifting the bag. Samples from each plot were placed in a sterile plastic bag and stored on ice (~4 °C) for transport back to USDA Genetics and Sustainable Agriculture Unit in Starkville, MS. A portion of each soil sample was separated for soil moisture content analysis. Soil moisture was assessed using the gravimetric method: 10 g of field-moist soil was weighed, followed by drying at 104 °C for 24 h and reweighing. Approximately 5 g of field-moist soil was separated and sent to Mississippi State University Soil Testing Lab to be analyzed for %OC using an automated dry combustion Vario Max Cube Elementar C/N analyzer (Elementar Americas Inc, Mt. Laurel, NJ). The remaining soil sample was transferred to a 473 mL (1 pint) container and sent to Pennsylvania State University Soil Fertility Lab Soil for analysis of pH and cation exchange capacity (CEC) by the Mehlich 3 (ICP) test.

## 2.4. $CO_2$ Flux

Carbon dioxide flux was determined for each plot using a LI-COR LI-8100A soil flux system. One PVC collar (20 cm diameter) was installed in the middle of each plot by inserting the collar into the soil leaving approximately 5 cm of the collar above the surface. Collars were leveled at insertion. The collars were established in the center of the plot within the planting bed. Four-point collar height measurements were taken from the soil surface to the top of the collar before each $CO_2$ flux reading to determine the chamber volume. Final $CO_2$ flux was adjusted for moisture, temperature and chamber volume using program SoilFluxPro (Version 4.0.1) [30–32]. Measurements were taken once a month between 9:00 AM–11:00 AM, beginning five days after planting and continuing throughout growing season (approximately May, June, July, August and September 2019 and 2020).

## 2.5. Bacterial and Fungal Abundance

Quantitative qPCR can be used as an indicator of soil biological properties [33]. Total microbial DNA was extracted from soil samples (0.25 g soil) using MP Biomedical FastDNA Spin Kits on a FastPrep-24 homogenizer. DNA was quantified and subjected to serial dilution(s) (10- to 100-fold) prior to polymerase chain reaction (PCR) assays to avoid inhibitory co-contaminants. Quantitative PCR (qPCR) assays for 16 S rRNA (16 S) [34–36] and 18 S rRNA [37] were conducted using previously identified cycling and reaction conditions. An Applied Biosystems StepOnePlus Real-Time PCR system was used in all qPCR assays. All qPCR assays were conducted in duplicate using ABI PowerSybr PCR Mix (Applied Biosystems), amplification signals were quantified with a PCR-based standard curve and confirmed using melt-curve analysis; controls included negative-template and inhibition controls. Samples were only recorded as positive when both duplicates exhibited amplification and melt-curve signal. Quantitative PCR results were corrected for soil moisture and reported as genomic units per g (GU g$^{-1}$).

## 2.6. Environmental Conditions

Hobo sensors were installed within 4 treatments plots (NT-cover, NT-NC, RT-cover and RT-NC). To account for soil type differences, hobo sensors were strategically placed in the second and the last block (Figure S1) where a plot with cover was next to NC under the same tillage treatment (Figure S1; plot 7–8 and 23–24). Because both treatments contained a rye CC, temperature differences between RY and RC were unlikely [38], therefore temperature

treatments are referred to as cover or no cover. Soil temperature and %RH was measured at 15 s intervals beginning in May through September in 2019 and 2020, when they were removed at the last sampling event of the year, prior to harvest. Sensors for %RH were installed in June of 2019, thus data for May 2019 were unavailable. Soil moisture was determined from each sample using the gravimetric method.

## 3. Statistical Analysis

All statistical analysis were conducted in Program R [39]. Prior to analysis, soil parameters were checked for general multivariate frequentist test assumptions including homogeneity of variance, multivariate normality and outlier detection. Generalized linear mixed models (GLMMs) were used to assess the relationship between management (CC and tillage regime), month and year on %OC, $CO_2$ flux, CC biomass, soil moisture and soil temperature, respectively. The interaction of tillage (NT or RT), CC (RY, RC, or NC), month and year were fixed effects and block was used as a random effect to account for the gradient of soil types observed in the study plots. Each model was checked for goodness of fit (homogeneity of variance, normality of residuals, residual distribution, normality of random effect and collinearity). Tukey's HSD *post hoc* test was used to identify pairwise differences between treatments. Alpha level was defined at 0.05. Marginal differences were considered at *p*-values 0.05–0.10.

In order to assess the relationship of soil and environmental conditions on %OC and $CO_2$ flux, respectively, a GLMM was fit to the data. We considered bacterial abundance (16 S), fungal abundance (18 S), soil moisture, soil temperature, cation exchange capacity (CEC), and pH as potential explanatory effects because they are often included in processed-based models of carbon sequestration [7,21,40,41]. In checking for multicollinearity, 16 S was found to be significantly positively correlated with 18 S ($r = 0.55$, $p < 0.05$) and moisture ($r = 0.63$, $p < 0.05$). Parameters pH and CEC also were found to have significant positive correlations ($r = 0.17$, $p < 0.01$). Therefore, 16 S, pH and temperature were the only variables used as fixed effects, and block was used as a random effect. Parameter estimates with 95% confidence intervals (CI) were calculated to assess the effect size. Covariates were centered and scaled prior to analysis.

Multivariate metrics were used to model C turnover in treatment groups along a temporal gradient. Prior to analysis, the data were scaled to be within range bounds using the "preProcess" command in package 'caret' [42]. Analysis was conducted in program R using package 'codyn' [43] and 'vegan' [44]. Change in the proportion of CC biomass, %OC and $CO_2$ flux between treatments was described in a time-lag analysis using Euclidian distances, calculated on pairwise C pools across 6 time points (May 2019, July 2019, September 2019, May 2020, July 2020 and September 2020). Because measurements of CC biomass were only available for 3 months (May, July and September), June and August were additionally omitted from the %OC and $CO_2$ flux dataset. Distance values were used in a regression against time intervals. The slope of the relationship describes the directional rate of change in C pools within each treatment. The number of points in the regression is the number of sample pairs and are therefore pseudo-replicated. Thus, the interpretation was on the shape and slope of the pattern, rather than the significance of the regression line. To visually explore the change in pools of C over time, relative abundance of %OC, $CO_2$ flux and CC biomass was calculated within each time point and treatment.

Using the relative abundances calculated above, the ratio of %OC to $CO_2$ flux (%OC: $CO_2$) was determined and then a GLMM was fit using cover and tillage as fixed effects and block as a random effect. Tukey's *post hoc* pairwise comparisons was used to identify pairwise differences in %OC: $CO_2$ between treatments. The full data set (all sampling time points) were used in the analysis to balance comparisons between %OC: $CO_2$ and %OC.

## 4. Results

### 4.1. Environmental Conditions

Mean soil temperatures for 2019 and 2020 showed significant differences in treatments, months and years ($F_{39,143130}$ = 1203, $R^2$ = 0.24, $p$ < 0.05; Figure 1A). Tukey's *post hoc* comparisons showed the overall temperature in cover to be significantly lower than no cover ($p$ < 0.01) and NT to be significantly greater than RT ($p$ < 0.01), although these differences varied across time (Figure 1A). The slope of the trend of soil moisture was positive in NT, May, September, 2020, RCxJuly, RCxJune, Septx2020 and Julyx2020. The slope of the trend of soil moisture was negative with Mayx2020 [45]. Tukey's *post hoc* comparisons showed significantly higher moisture in 2019 than 2020 and higher in NT than RT ($p$ < 0.001; Figure 1B) but was not different between covers treatments. Percent relative humidity was significantly different between months and years ($F_{8,66650}$ = 619, $R^2$ = 0.06, $p$ < 0.01; Figure 1C).

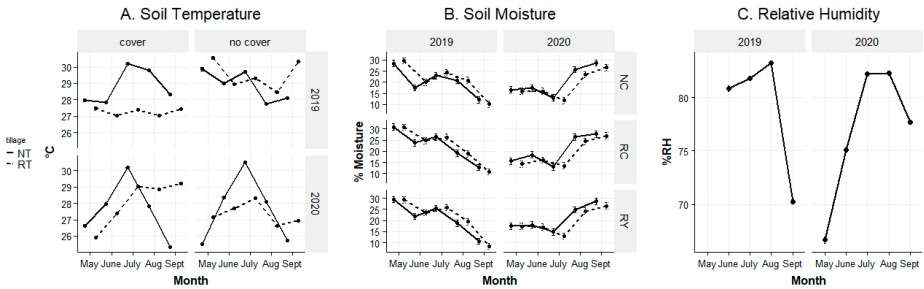

**Figure 1.** Soil Temperature, Soil Moisture and Relative Humidity, soil temperature (**A**), soil moisture (**B**), and relative humidity (**C**).

Changes in soil temperature (A), soil moisture (B), and relative humidity (C) in each treatment (NC = no cover, RC = rye + clover cover crop, RY = rye cover crop; NT = no till, RT = reduced till) over the growing season (May, June, July, August (Aug), September (Sept) during the study years (2019 and 2020). Treatment plots located were located in Stoneville, MS, USA.

### 4.2. Soil Organic Carbon

No-till had a positive effect on %OC as well as both covers (RC and RY) in the month of September ($R^2$:0.51; Table S1). Tukey's pairwise comparisons revealed that %OC was significantly higher in NT (1.27% ± 0.03) than RT (1.10% ± 0.03; $p$ < 0.01) and greater in both CC (RY: 1.23% ± 0.03, RC:1.22% ± 0.03) than NC (1.11% ± 0.03; $p$ < 0.01; Table 1, Figure 2). There were marginal differences between 2019 and 2020 ($p$ < 0.07; 2019: 1.2%, 2020: 1.15% ± 0.02). When assessing %OC relationships with soil characteristics, bacteria abundance had a significant positive effect on %OC, while pH had a significant negative effect ($R^2$:0.97; Table S2; Figure 3).

**Table 1.** Predicted %OC within treatments.

| Tillage | Cover | Mean | 2.5% CI | 97.5% CI | Sig Difference |
|---------|-------|------|---------|----------|----------------|
| NT | NC | 1.21 | 1.11 | 1.31 | A |
| | RC | 1.31 | 1.21 | 1.41 | A |
| | RY | 1.28 | 1.18 | 1.38 | A |
| RT | NC | 1.01 | 0.91 | 1.11 | B |
| | RC | 1.12 | 1.02 | 1.22 | A |
| | RY | 1.17 | 1.07 | 1.27 | A |

Model predicted values of percent organic carbon (%OC) with 95% confidence intervals (CI) within study treatments (NT = no-till, RT = reduced till, NC = no cover, RC = rye + clover cover crop, RY = rye cover crop) in study plots located in Stoneville, MS, USA. Significant differences (Sig difference) in Tukey's *post hoc* pairwise comparisons are denoted by lack of a shared letter.

## Percent Organic Carbon

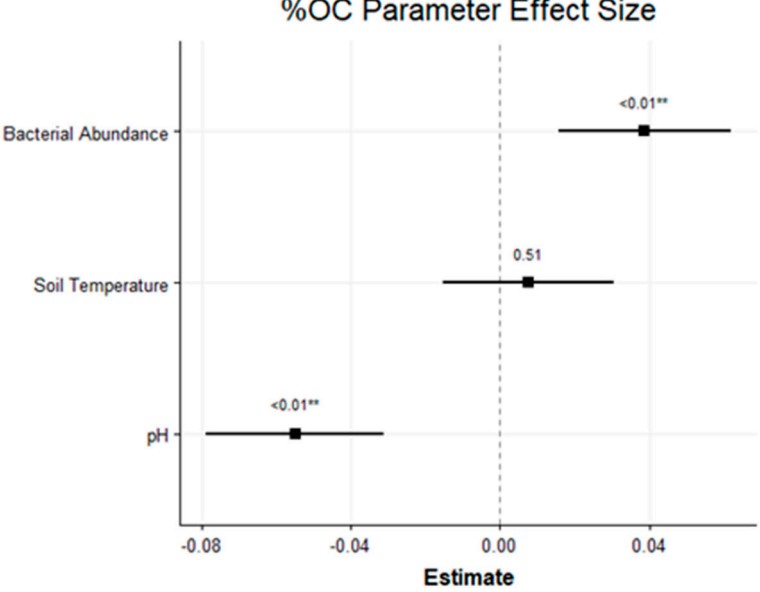

**Figure 2.** Percent Organic Carbon within Treatments.

## %OC Parameter Effect Size

**Figure 3.** Percent Organic Carbon Parameter Effect Size. Double asterisk (**) denotes a significant effect.

Model predicted values of percent organic carbon (%OC) measured over growing season of study years (May-September 2019 and 2020) within each combination of treatments (NT = no-till, RT = reduced till, NC = no cover, RC = rye + clover cover crop, RY = rye cover crop). Error bars denote standard error. Treatment plots were located in Stoneville, MS, USA.

Parameter (bacterial abundance, soil temperature and pH) effect size on percent organic carbon (%OC) in study plots located in Stoneville, MS, USA. Solid lines denote 95% confidence intervals. *p*-values are reported above confidence intervals in each corresponding parameter. Double asterisk (**) denotes a significant effect.

### 4.3. CO$_2$ Flux

No-till and the interaction of May, July and Septx2020 had a positive effect on CO$_2$ flux, while May, June, July, September and 2020 had a negative effect ($R^2$:0.49, Table S3). Tukey's pairwise comparisons revealed no significant differences between tillage or CC treatments, however CO$_2$ flux was significantly higher in 2020 ($p < 0.001$; 481 mg CO$_2$ m/h $\pm$ 11.1) than 2019 (387 mg CO$_2$ m/h $\pm$ 11.1). Carbon dioxide flux was also significantly different between months ($p < 0.03$; Figure 4) with highest emissions occurring in July (492 mg CO$_2$ m/h $\pm$ 13.1) and lowest emissions in September (377 mg CO$_2$ m/h $\pm$ 13.1). There were marginal differences between NTxRY and RTxRY in August of 2019 and between NT and RT in all cover treatments in July of 2020 ($p = 0.11$; Figure 4). When assessing CO$_2$ flux relationships with soil characteristics, bacterial abundance and pH had a significant negative effect on CO$_2$ flux ($R^2$:0.96; Table S2; Figure 5). Bacterial and fungal abundances are depicted in Figure 6.

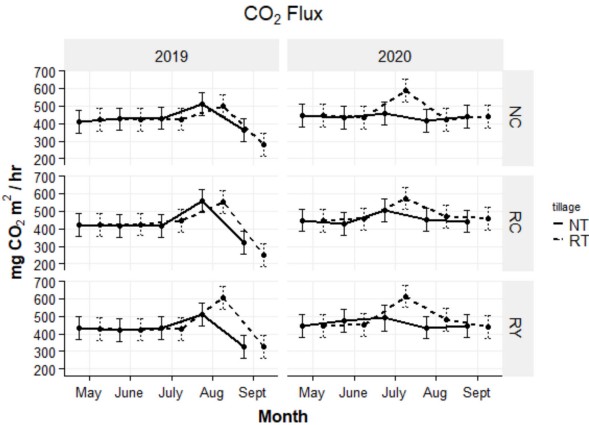

**Figure 4.** CO$_2$ flux within treatments.

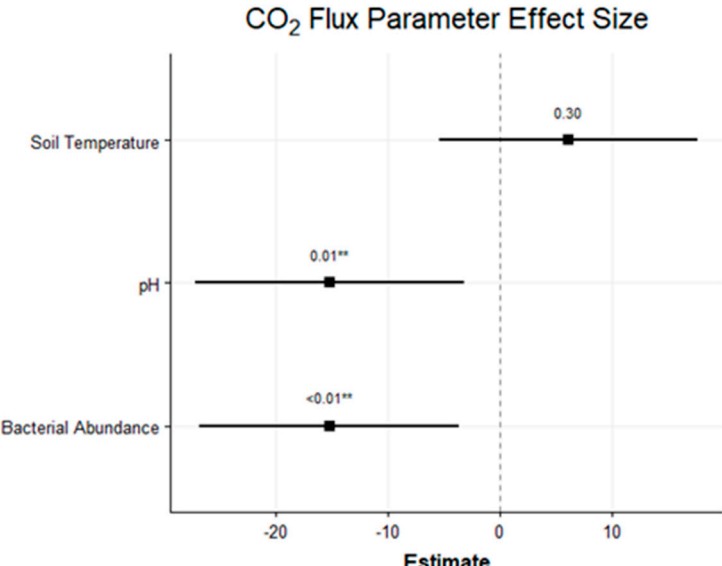

**Figure 5.** CO$_2$ Parameter Effect Size. Parameter (soil temperature, pH, bacterial abundance) effect size on carbon dioxide (CO$_2$) flux in study plots located in Stoneville, MS, USA. Solid lines denote 95% confidence intervals. *p*-values are reported above confidence intervals in each corresponding parameter. Double asterisk (**) denotes a significant effect.

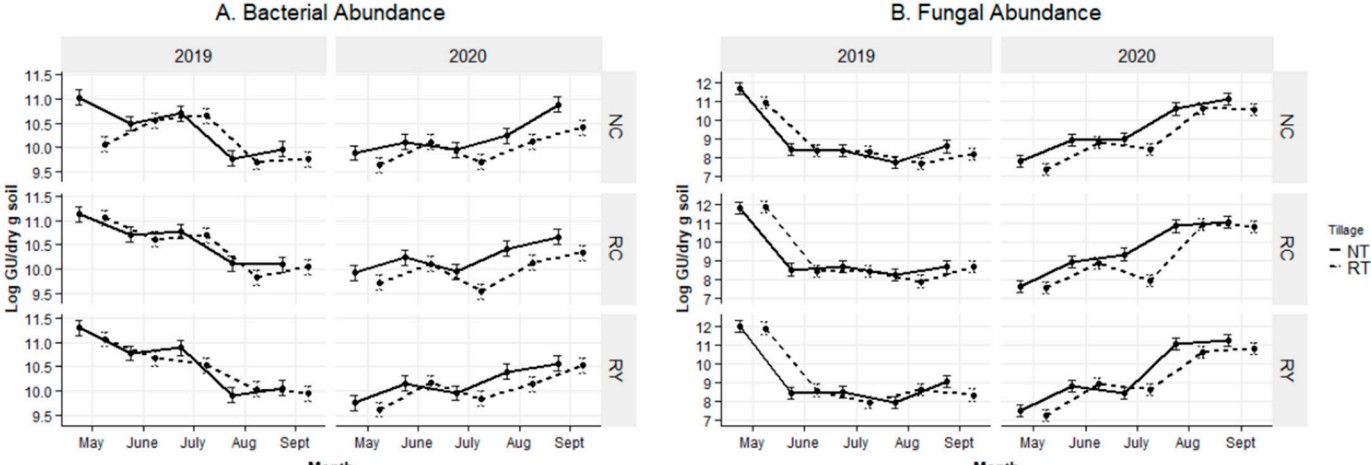

**Figure 6.** Bacterial and Fungal Abundance. Model predicted values of Bacterial (**A**) and Fungal (**B**) abundances in each treatment combination within each combination of treatments (NT = no-till, RT = reduced till, NC = no cover, RC = rye + clover cover crop, RY = rye cover crop) over time (May–September 2019 and 2020) reported in [45]. Error bars depict standard error of predicted values. Treatment plots were located in Stoneville, MS, USA.

Model predicted values of carbon dioxide ($CO_2$) flux measured over growing season of study years (May-September 2019 and 2020) within each combination of treatments (NT = no-till, RT = reduced till, NC = no cover, RC = rye + clover cover crop, RY = rye cover crop). Error bars denote standard error of predicted values. Treatment plots were located in Stoneville, MS, USA.

*4.4. CC Biomass*

Rye and RC had a positive effect on CC biomass. There was a marginal positive effect of RCxMay and negative effect of RCxSept and RCxSeptxYear. The YearxCC interaction also had a negative effect ($R^2$:0.84, Table S4). Tukey's pairwise comparisons revealed significantly more CC biomass in 2019 (309 g/mg$^2$ ± 6.01) than in 2020 (197 g/mg$^2$ ± 6.01). Rye + clover CC had the greatest CC biomass (366 g/mg$^2$ ± 12.4) followed by RY (285 g/mg$^2$ ± 6.01) and NC (110 g/mg$^2$ ± 12.4). In 2020, there was substantial volunteer CC biomass in the NC plots, but still significantly less cover than the intended CC plots. There was no significant difference in CC biomass when comparing biomass in NT and RT.

*4.5. Directional Change of C Turnover*

Carbon pools and temporal variability showed variability among treatment groups (Figure 7). Plots that included a cover crop (RY and RC) showed a more positive directional change than those without (Table 2, Figure 7), that is, the rate of change ($b$ = slope) increased with the addition of a CC (Table 2), indicating a reordering pattern of C pool abundance. Directional change was reflected in the relative abundance of each C pool over time (Figure 8). NTxRT and NTxRY displayed temporal changes favoring a dominate %OC pool, meaning that the relative abundance of %OC was greater than the relative abundance of $CO_2$ flux. All RT treatment combinations suggested a dominating $CO_2$ flux pool (Figure 8). NTxNC showed no change in C pools (Figure 8) which is consistent with a neutral slope ($b$).

**Figure 7.** Directional Change of Carbon Pools. Time lag regressional analysis of composition change in carbon (C) pools on different treatment combinations treatments (NT = no-till, RT = reduced till, NC = no cover, RC = rye + clover cover crop, RY = rye cover crop). Differences in C composition are characterized by Euclidean distances, which are calculated on pair-wise C pools across the entire time series intervals (May, July, September 2019 and 2020). Plots that included a cover crop (RY and RC) showed stronger directional change than those without (NC). Positive slopes indicate that early time points of C pool composition are more dissimilar to time points further away than those closer together, a negative slope indicates that early time points are more similar to points far away than close, and a neutral slope indicates a no pattern/change in dissimilarity. Treatment plots were located in Stoneville, MS, USA.

**Table 2.** Rate of Directional Change in C Pools.

| Tillage | Cover | *b* |
|---------|-------|-----|
| NT | NC | 0.0229 |
|  | RC | 0.23 |
|  | RY | 0.169 |
| RT | NC | −0.16 |
|  | RC | 0.213 |
|  | RY | 0.165 |

Rate of directional change in C pools in study treatments (NT = no-till, RT = reduced till, NC = no cover, RC = rye + clover cover crop, RY = rye cover crop). Slopes (*b*) of regression time-lag analysis indicate C pools reordering over time. Treatment plots were located in Stoneville, MS, USA.

Tillage and RC had a significant positive effect on $\%OC:CO_2$ ($R^2c = 0.78$; Table S5). Tukey's pairwise comparisons showed RT to be significantly lower than NT ($p < 0.01$), which suggests %OC loss. Both CC treatments showed significantly higher ratios than NC ($p < 0.01$), suggesting %OC accumulation in plots with cover, which is consistent with the directional change of C pools. When comparing the interaction of tillage and CC, $\%OC:CO_2$ showed NTxRY and NTxRC to be significantly greater than RTxRC ($p < 0.01$). The same relationship was not observed when examining differences in %OC alone. Likewise, there was no significant difference between $\%OC:CO_2$ RTxRC and RTxNC ($p = 0.25$), but a significant difference was observed when comparing the treatments %OC (Table 1). Predicted $\%OC:CO_2$ ratios can be found in Table 3.

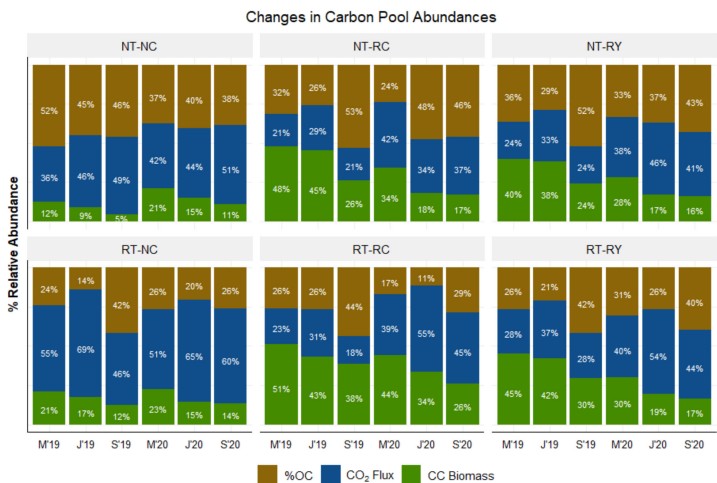

**Figure 8.** Changes in Carbon Pool Abundances. Figure 8 depicts changes in relative abundances (percentages shown) of each carbon (C) pool (percent organic carbon (%OC), carbon dioxide ($CO_2$) Flux and cover crop (CC) biomass) within each treatment combination (NT = no-till, RT = reduced till, NC = no cover, RC = rye + clover cover crop, RY = rye cover crop) at each month (M = May, J = July, S = September) in the study years (19 = 2019, 20 = 2020). Treatment plots were located in Stoneville, MS, USA.

**Table 3.** Predicted %OC:$CO_2$ within Treatments.

| Tillage | Cover | Mean | 2.5% CI | 97.5% CI | Sig Difference |
|---------|-------|------|---------|----------|----------------|
| NT | NC | 0.98 | 0.64 | 1.32 | ABab |
|    | RC | 1.29 | 0.95 | 1.63 | Bb |
|    | RY | 1.14 | 0.80 | 1.48 | ABb |
| RT | NC | 0.48 | 0.14 | 0.82 | aC |
|    | RC | 0.78 | 0.44 | 1.12 | AaC |
|    | RY | 0.87 | 0.53 | 1.21 | AB |

Model predicted values of the ratio of percent organic carbon to carbon dioxide flux (%OC:$CO_2$) with 95% confidence intervals (CI) within study treatments (NT = no-till, RT = reduced till, NC = no cover, RC = rye + clover cover crop, RY = rye cover crop) in study plots located in Stoneville, MS, USA. Ratios were determined from estimated relative abundances. Significant differences (Sig difference) in Tukey's post hoc pairwise comparisons are denoted by lack of a shared letter and letter case.

## 5. Discussion

### 5.1. Management Practice Effects on %OC and $CO_2$ Flux

A %OC increase in plots managed with NT and CC is consistent with other studies from the region [26,46], where %OC increased in all plots that included CC. Cover crops provide an additional carbon source to the cropping system, increasing %OC as CC biomass degrades, echoed in the 3.5–27.3% difference in OC g kg$^{-1}$ in study plots that included a CC compared to plots without. In a regionally comparable study of CCs conducted in Louisiana, USA, Sanchez et al. [46] looked at the effects of CC and a variable N fertilizer rate on %OC. They found a 35% increase in %OC g kg$^{-1}$ after only two years, suggesting the degree of %OC increase will be influenced by other factors, not just time (i.e., additional management, climate and edaphic properties). In a recent article by Daryanto et al. [19], ecosystem services of CC were quantitatively synthesized. It was found that CC consistently increase %OC concentration in soil by at least 9%, regardless of climatic region and management, indicating that CC are a positive practice for %OC accumulation under most agronomic conditions.

Soil organic carbon is highly dynamic and its residence time in soil will depend on soil biological and physical properties as well as climate conditions [47]. In this study, %OC was negatively affected by soil pH and positively affected by bacterial abundance. Generally, %OC and pH have a negative relationship, which has been documented at

various landscape scales [23,48]. When organic matter decomposes (e.g., CC residue), organic acids are released which lowers pH. Additionally, slight increases in pH can activate bacterial populations that are dormant at a lower pH, resulting in decomposition of organic matter and a %OC loss. Soil bacteria (and other microorganisms) contribute to the degradation and mineralization of organic matter as well as the to the generation of new organic metabolites. The produced metabolites form strong chemical bonds with the mineral soil matrix, increasing stable soil C [41,49,50]. Counter to most statistical models that use soil temperature to predict %OC [7,51], temperature was not a significant predictor of %OC in the study plots. This may be because soil temperatures were mean values over a month period, which limits predictive power. Future studies should incorporate continuous temperature and measurements over time to fully capture their influence over %OC in the study system.

Studies of CC influence on $CO_2$ flux generally predict increased GHG emissions because of increased decomposition of organic matter by soil microbes. However, no overall difference in $CO_2$ flux was found between CC treatments. Romdhane et al. [52] found that management of CC had a larger influence on soil microbial dynamics than the addition of CC alone. The degree that CC residue comes into contact with soil through rolling, tilling, or standing, will shift rates of decomposition and mineralization. We hypothesize that significant differences in $CO_2$ flux were not observed because there was minimal CC contact with soil after termination and there was substantial volunteer vegetation growth in NC plots. Changes in $CO_2$ flux between treatments may be observed post-harvest, when RT treatments incorporate residue, therefore future studies should include $CO_2$ flux measures for a full year.

While overall there was no difference in $CO_2$ flux between treatments, $CO_2$ flux was lower in NT in July of 2020 when compared to RT. Percent relative humidity was also high that month, which has been linked to faster decomposition rates and an increase in $CO_2$ flux [53]. Interestingly, bacterial abundance had a negative effect on $CO_2$ flux, which was reflected in greater abundances of bacteria and fungi in NT plots in July of 2020. This inverse relationship is counter to most studies where an increase in microbial activity is generally associated with higher $CO_2$ emissions [18,19,54]. We postulate two reasons for this: (1) In July 2020, soil temperature spiked and soil moisture declined, potentially triggering a stress response in soil microorganisms, thereby increasing the proportion of metabolically dormant bacteria. Dormant bacteria do not respire. Since bacterial and fungal abundance was measured as rRNA detection, abundances reflect presence of microorganisms, not microbial activity. (2) No-till promotes microbial communities that employ anaerobic respiration. In Firth [45] it was found that NT plots had different bacteria community composition and lower diversity than RT which was attributed to low $O_2$ conditions in heavy clay soils. Because only $CO_2$ flux was measured, we cannot determine if a microbial community shift towards facultative anaerobes that favor different processes of alternative terminal electron acceptors occurred. This would potentially result in increased methane or nitrous oxide emissions. It's likely that both postulates are contributing to the observed relationship of $CO_2$ flux and microbial abundance. We conclude that NT management on MAV clay soils may help mitigate $CO_2$ flux when %RH is high, but there is considerable uncertainty in the mechanism and alternative outcomes.

*5.2. Directional Change of C Turnover*

Measures of C turnover are more sensitive to temporal shifts in C pool composition than when assessing each pool individually. When examining directional change of C (or turnover), a positive slope, as seen in treatment combinations that included CCs (RY or RC), indicates that there are proportional shifts in C pools that are directional, that is, measured pools are reordering themselves through time towards compositions that are increasingly dissimilar. Logically, the decomposition of an additional C resource will be differentially affected by climate conditions and resident soil microbial communities and thus rearrange C pools. While this is a somewhat obvious conclusion, the positive slopes

confirm that directional change is heading towards a dominate pool. Visualizations of relative abundances show RT-CC combinations favoring $CO_2$ flux and NT-CC combinations favoring %OC and CC biomass (Figure 8). The negative slope in RTxNC also confirms a pattern in variable change, but one of stability, meaning that the relative abundances of C pools cycle back to their original state towards a dominate $CO_2$ flux pool. Interestingly, the neutral slope in NTxNC suggests no pattern in C turnover, or rather relative abundances do not change.

It is important to note that directional change towards %OC accumulation does not mean C stabilization. Changes in environmental conditions, for example increased temperatures, rainfall, or nitrogen addition, can quickly change the rate and mechanisms of %OC decomposition [40,55]. Therefore, if accumulated %OC is in a labile form, it is still vulnerable to system loss if environmental conditions were to shift. Nevertheless, physiochemical traits of soil (clay fraction, CEC, and pH) are an equally important controlling factor towards C sequestration and stabilization [7,21,56]. Likewise, recalcitrant organic chemicals, such as soybean-derived phytotoxins, are also bound by soil particles and contribute to soil organic carbon [57]. Cation exchange capacity is associated with available surface area of soil which reflects the sorption potential of %OC into soil aggregates, thereby protecting %OC from decomposition [58]. After controlling for the clay soil fraction, CEC was found to be significantly higher in NT plots than those without [45], which is encouraging that at least a fraction of accumulated %OC in NT-CC plots may be retained in a stable form.

The advantage of this approach is that trends towards C loss or C accumulation can be identified over short time periods with fewer parameters. The utility of C turnover lies in its ability to be swiftly translated into a metric evaluating conservation practice performance towards climate mitigation objectives. However, we recognize that only accounting for %OC, $CO_2$ flux and CC biomass neglects several important C pools, namely above-and-below ground plant biomass and methane. The inclusion of methane is particularly important for interpretation of this study because of the postulate that NT treatments are shifting microbial communities towards facultative anaerobes. Including all C pools may result in a shift towards net C loss in those treatments that currently favor a dominating %OC pool (or %OC accumulation). Additionally, our results only account for growing season. The proportional changes will likely fluctuate if fallow season measurements are included. However, we reason that while the relative abundances of %OC, $CO_2$ flux and CC biomass will change with the incorporation of more C pools and time measurements, the relative ratios will remain the same and therefore still convey important information about C relationships, albeit incomplete.

To illustrate this point, we compare measures of %OC to %OC:$CO_2$. Soil organic carbon was greater in RTxRC than RTxNC, but there was no difference between these groups when looking at %OC:$CO_2$ (Tables 1 and 3). If the management objective was to accumulate %OC in a RT system, RC would increase input costs but may not result in a detectable change in %OC over time. Likewise, NTxRY and NTxRC %OC was not significantly different than RTxRC. However, RTxRC %OC:$CO_2$ is significantly lower than NTxRY and NTxRC (Tables 1 and 3), meaning that despite having a similar %OC average, RTxRC is losing more %OC from the system than NTxRY and NTxRC. This is an important observation for a producer that could result in a significant economic return which would otherwise not been apparent if looking %OC independently. We encourage future studies to incorporate a more comprehensive assessment of major C pools to increase confidence in identified trends of C turnover and facilitate the translation of findings to management action.

## 6. Conclusions

Landscape-scale management directives for C storage are receiving global attention and at the center of national policy discussion. Producers are challenged to adopt conservation practices where successful implementation is often dependent on local weather patterns, land history and site-specific features that drive crop production and success. In this study we addressed changes in C following long-term application of CC and NT

management in a soybean cropping system in the mid-South. Conservation management increased %OC, showed no overall difference in $CO_2$ flux between treatments, and CC biomass increased with the number of species in the CC mix. Soil organic carbon was positively affected by increases in microbial abundance, while $CO_2$ flux was negatively associated with microbial abundance. We hypothesize that the combination of environmental conditions and lower $O_2$ soils contributed to a decrease in microbial activity and a shift towards facultative microbial communities that favor alternative terminal electron acceptors, in turn lowering $CO_2$ respiration. Carbon dioxide flux was also associated with increases in %RH, supporting Thapa et. al.'s [53] notion that processed-based models of decomposition should be updated to include %RH.

Finally, conceptualizing C as proportional pools that fluctuate over time is a more holistic assessment of C dynamics that is easily transferred to public understanding. Using a measure of C turnover to evaluate compositional changes in C pools shows trends towards incremental shifts in C loss or accumulation. In this study, measures of C turnover suggested that NT combined with CC resulted in C accumulation, while RT combined with a CC suggested C system loss which was not apparent when assessing %OC alone. Future research should assess how other GHGs (i.e., methane, nitrous oxide) are affected by heavy clay soils in the mid-South, USA, as well include above and below-ground plant biomass as C pools in a directional model. Trends in C turnover can then be more confidently identified and translated into swift management action.

**Supplementary Materials:** The following supporting information can be downloaded at: https://www.mdpi.com/article/10.3390/environments9090109/s1, Figure S1: Treatment plot map. Table S1: Percent Organic Carbon Mixed Model Results, Table S2: $CO_2$ Flux Mixed Model Results, Table S3: Mixed Model Results of Soil Environmental Variables Relationship with %OC and $CO_2$ Flux, Table S4: Cover Crop Biomass Mixed Model Results, Table S5: %OC to $CO_2$ Flux Mixed Model Results.

**Author Contributions:** Conceptualization, A.G.F., M.A.L. and B.H.B.; methodology, A.G.F., J.P.B., M.A.L. and B.H.B.; formal analysis, A.G.F.; investigation, A.G.F.; resources, J.P.B., M.A.L. and B.H.B.; writing—original draft preparation, A.G.F.; writing—review and editing, A.G.F., J.P.B., M.A.L., D.J.M., A.B. and B.H.B.; supervision, J.P.B., B.H.B.; funding acquisition, J.P.B., M.A.L. All authors have read and agreed to the published version of the manuscript.

**Funding:** Research support was provided to Mississippi State University through USDA ARS Agreement 58-6001-8-003.

**Data Availability Statement:** The data presented in this study are available on request from the corresponding author.

**Acknowledgments:** Heather Tyler, Krishna Reddy, and technical support staff of the USDA-ARS Crop Production Unit, in Stoneville, MS, for maintaining the research plots and facilitating the research. Nicky Faucheux for her guidance and support in the development of the manuscript.

**Conflicts of Interest:** The authors declare no conflict of interest.

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
