# Peer review of "Dynamics of Soil Organic Carbon and CO2 Flux under Cover Crop and No-Till Management in Soybean Cropping Systems of the Mid-South (USA)"

_environments, doi:10.3390/environments9090109_

Round 1
Reviewer 1 Report
Manuscript title: Dynamics of soil organic carbon and CO₂ flux under cover crop and no-till management in soybean cropping systems of the mid-South
Manuscript id: environments-1813613
Authors: Firth et al.
The manuscript regarding the topic and results presented is of interest to environmental science community and revisions based on the comments below are recommended before considering for publication.
Major comments
· I suggest to add ‘’ United States’’ after ‘’ mid-South’’ to help the readers on the location
· The unit / abbreviation is not mention before, consider define the abbreviation when mentioned for the first time…. Please check throughout the manuscript to define the abbreviations.
· Lake of scientific literature to support the statements and finings throughout the manuscript…... I have made a few suggestions for that, and more need it….
· More information needed for ALL TABLE captions and define the abbreviation and units that used. And adjust the significant figures for the table and manuscript.
· I have a major concern about the results and discussion section. The authors describe results and compare the results with previous studies, however, insight mechanisms are still not sufficient.
Minor comments:
Abstract
The unit / abbreviation is not mention before, consider define the abbreviation when mentioned for the first time.
Introduction:
Line 41-43: A reference needed here, and the same for line 48-53
Line 82: Please add ‘’ United States’’
Line 97-100: Is this statement is part of the aims for this study? If yes, there is no linkage with the following the linkage between
In MM section
Literature references are missing for all sub-section. It would be better to cite the references that the procedure adapted.
With section 2.1 a graph or illustration would help very much to visualize the field experiment. In addition, any fertilizer applied to the field ? if yes, please elaborate!
Please spell out the stat abbreviations, where the experiment conducted. e.g MS in line 160
Line 145-150: Consider writing the equation too.
In general. how many times you’ve recorded the data,? duplicate? Triplicate?....., please elaborate more on this
R&D section
Do have the precipitation data? I believe precipitation is one the major factors that affected such significant changes with 2 years, for example in Fig 1B, 1C, Fig 2, Fig 4 RY,
Figure 1:B: do we know why the data is completely different between 2019 and 2022?
These sections are repeating information already presented and explaining things in an unnecessarily complicate way. The quality of the manuscript would benefit from the whole section being condensed, Line 278-289, line 348-357, Line 405-436……
It would be great to emphasize that the microbial community pattern is not significantly difference among the time, which is really interesting how other parameters (which some of them related to nutrition in the soil) have not promoted their diversity in the soil……
Figure 7B; Month labelling is not clear, please consider highlight the months inside the graphs.
Line 453-455: Line 28-31: consider using this reference: https://doi.org/10.1021/acs.est.1c01477
Conclusion
Nice conclusions! However, the future perspectives for following research highly crucial here …..
Reviewer 2 Report
The present study gives important inights on the role of cover crop and tillage managements on soil organic carbon because it took in consideration its inputs, CO2 emissions from soils and soil microorganisms.
The manuscript is well organized and just a minor English spell checking is required. Some more comments are reported below
In the title, add USA between brackets
L 35: 60 % in one year? please report the time span
L 36: what do you mean for "extract"? Maybe you meant that the poor soil practices promote organic carbon mineralization
Ls 38-40: In my opinion it is enough to report the potential of soils to store C which must be seen as an effective strategy for climate change mitigation
L 35: please report the time span
Ls 62-63: Is it necessary to report the yield reduction due to the presence of rye as CC? In your manuscript, the yield is not considered. I suggest you to remove it.
L 89: what do you mean for "complex interaction between SOC and soil mineral phases"? In any part of the world there are complex interactions between SOC and soil mineral particles
Ls 89-92: this part can be removed
L 114: which blocks?
L 118: since spring of 2008
L 130: this paragraph could be moved after the paragraph 2.6 of the present version of the manuscript
L 131: "cover" is too vague, please report in which plot the sensors were installed
L 139: can you report the most abuntant plant species found in NC plots?
L 141: you can just say that the plant biomass was dried at 105°C and the dry mass weight was measured
L 145: when did you secure litter bags on soil surface?
L 145: it is not clear why you prepared 48 litterbags instead of 24 (one for each plot)
L 152: the other one?
L 154: finally, how many samplings did you perform every year?
Ls 157-158: if you secured the litterbags on soil surface, how did you collect soil samples directly under the litterbag?
L164: here and elsewhere within the text, use SOC content instead of %OC
L 205: Alpha level
Ls 240-241: please, write some words about these differences
L 244: at Ls 241-243 you reported that 2020 had a positive effect on soil moisture which means that in 2020 you had higher moisture content compared to 2019. Here, you report that in 2019 you found higher soil moisture compared to 2020. Please check. However, I suggest to rewrite Ls 241-243 to better understand the positive and negative effects that you reported
Figure 1: the figure caption must be rewritten, it does not help to understand the Figure. For example, some abbreviations are not explained
L 253: written in the present form, it sound like to till soil promote SOC accumulation. Probably you meant that tillage had a significant effect on SOC content
Ls 258-260: these two sentences can be removed
L 263: why did you use "predicted"? You measured SOC content
Table 1: This Table is not self explanatory. For example what is the meaning of CI? What do the letters of the last column mean? Please, all the Tables and Figure should be revised in order to be self explanatory
Ls 364-365: It is not necessary to say again the aim
Ls 365-374: You are in the Discussion section, therefore you have to explain your findings in order that the readers can understand the processes behind them otherwise it will sound as Result section
Round 2
Reviewer 1 Report
To Author
The revised manuscript has improved compared to the original version. I recommend the manuscript to be published!
Best wishes,